# Adenosine, Schizophrenia and Cancer: Does the Purinergic System Offer a Pathway to Treatment?

**DOI:** 10.3390/ijms231911835

**Published:** 2022-10-05

**Authors:** Abdul-Rizaq Hamoud, Karen Bach, Ojal Kakrecha, Nicholas Henkel, Xiaojun Wu, Robert E. McCullumsmith, Sinead M. O’Donovan

**Affiliations:** 1Department of Neurosciences, University of Toledo, Toledo, OH 43614, USA; 2Neurosciences Institute, ProMedica, Toledo, OH 43606, USA

**Keywords:** schizophrenia, cancer, adenosine, purinergic signaling, epidemiology

## Abstract

For over a century, a complex relationship between schizophrenia diagnosis and development of many cancers has been observed. Findings from epidemiological studies are mixed, with reports of increased, reduced, or no difference in cancer incidence in schizophrenia patients. However, as risk factors for cancer, including elevated smoking rates and substance abuse, are commonly associated with this patient population, it is surprising that cancer incidence is not higher. Various factors may account for the proposed reduction in cancer incidence rates including pathophysiological changes associated with disease. Perturbations of the adenosine system are hypothesized to contribute to the neurobiology of schizophrenia. Conversely, hyperfunction of the adenosine system is found in the tumor microenvironment in cancer and targeting the adenosine system therapeutically is a promising area of research in this disease. We outline the current biochemical and pharmacological evidence for hypofunction of the adenosine system in schizophrenia, and the role of increased adenosine metabolism in the tumor microenvironment. In the context of the relatively limited literature on this patient population, we discuss whether hypofunction of this system in schizophrenia, may counteract the immunosuppressive role of adenosine in the tumor microenvironment. We also highlight the importance of studies examining the adenosine system in this subset of patients for the potential insight they may offer into these complex disorders.

## 1. Introduction

Epidemiological observations dating back over one hundred years suggest that incidence rates of some cancers may be lower in patients with schizophrenia [1,2,3,4,5]. Numerous studies have been conducted in the decades since then but have failed to reach a consensus; there are reports of increased, reduced, or no difference in incidence rates of most cancers in schizophrenia patients compared to the general population (see reviews [2,3,4,5] and epidemiology studies [5,6,7,8,9,10,11]). However, as many of the risk factors for cancer, including smoking, alcohol use, increased body weight, physical inactivity, and reduced access to medical care, are also found in schizophrenia populations, it is surprising that cancer incidence rates are not higher amongst these patients [4,6,12,13,14,15,16,17]. As has previously been observed, this unexpected finding lends support to the idea that a diagnosis of schizophrenia may be protective against development of (some) cancers [18].

## 2. Cancer Incidence in Schizophrenia

A limitation of many epidemiological studies of cancer incidence in schizophrenia is that they do not control for cancer-related confounding variables. Age, for example, is a major confounding variable, as patients with schizophrenia typically have a shorter lifespan [19], making comparisons of cancer incidence with control populations a challenge. It is also crucial to precisely define control groups (general population, patients in inpatient care or outpatient care, non-schizophrenia, non-psychiatric, etc.) and to match study subjects based on age and sex. However, when risk factors like age and smoking are considered in analyses, the reported cancer incidence rates are lower than would be expected in this patient population [6,7]. It is also important to note that cancer incidence rates, and not cancer mortality rates, are proposed to be lower for some cancers in schizophrenia patients. Survival rates of schizophrenia patients who develop cancer are typically lower than the general public [20,21]. This is especially interesting when considering separate biological mechanisms regulate cancer initiation, proliferation and metastasis [22]. In Table 1 we summarize the results of recent epidemiology meta-analyses reporting cancer rates in schizophrenia patients. Overall, a limited number of prospective cohort studies were available for inclusion in these meta-analyses, missing medication history and detailed demographic information is common, variable data collection methods and the use of inconsistently defined control groups contribute to challenges in interpreting the data and determining whether cancer incidence rates are significantly reduced in schizophrenia patients [3,23]. Interestingly, breast cancer incidence rates are consistently higher in studies of schizophrenia populations (Table 1). Recent genome wide association studies (GWAS) have identified a shared risk locus associated with schizophrenia and breast cancer [24] and increased risk for breast, ovarian and thyroid cancers, but not other cancers, in schizophrenia patients [25]. Conversely, lung, liver and prostate cancer incidence appears to be lower in schizophrenia populations (Table 1), suggesting that incidence rates of some cancers may be reduced, or unchanged, in these patients. Additional appropriately powered studies that consider different cancer types and account for potentially confounding variables (age, BMI, smoking, medications, genetics, and geography) will be required to ascertain whether a diagnosis of schizophrenia protects against the development of some cancers.

Chronic antipsychotic use, genetic factors, and pathophysiological changes have been posited to explain the potentially reduced risk of cancer development in schizophrenia patients. For example, there is a growing body of evidence that antipsychotics may protect against the development of cancer; antipsychotics are now being repurposed as potential cancer therapies, with anticancer mechanisms of action in peripheral tissues still poorly understood [6,28,29,30,31]. Epidemiological studies have found that unaffected family members of patients with schizophrenia have reduced rates of cancer [4,32], suggesting that genetic factors contributing to schizophrenia risk may also reduce the risk of developing cancer. However, much less is known about how pathophysiological changes associated with schizophrenia may contribute to the reduced risk of cancer development in some patients [33,34].

The potential protective effects of the adenosine system against cancer development in schizophrenia patients was first suggested over 20 years ago in the “Purinergic Hypothesis of Schizophrenia” by Lara and Souza [35,36]. The authors noted the reduced rates of cancer in schizophrenia patients, and proposed a role for the purinergic system in this phenomenon, although little was known about the function of the adenosine system in cancer at that time [37,38]. Today, the purinergic hypothesis has evolved, with a growing body of evidence suggesting that reduced availability of extracellular adenosine, and the resulting effects on neuromodulation, and immune system and bioenergetic regulation, may contribute to the onset of schizophrenia [39]. Conversely, increased adenosine metabolism is now recognized as a characteristic of many types of cancer. Elevated adenosine, and its metabolic precursor adenosine triphosphate (ATP), promote cancer growth and play a role in host immunosuppression in the tumor microenvironment (TME) [36,40,41,42,43].

Herein, we discuss the biochemical and pharmacological evidence for hypofunction of the adenosine system in schizophrenia and hyperfunction in cancer (Figure 1). We draw primarily from postmortem and clinical studies to explore whether perturbation of the adenosine system may contribute to reduced risk for cancer development in some schizophrenia patients. We also review the literature showing that augmentation or attenuation of the adenosine system can have therapeutic benefits for these disorders.

## 3. The Adenosine System: Extracellular Adenosine Generating Pathways

The ratio of ATP:adenosine is tightly regulated by enzymatic pathways that modulate the generation of extracellular adenosine (Figure 2). Adenosine can be directly released from the cell by nucleoside transporters [44] where it is rapidly catabolized to inosine by the purine salvage pathway enzyme adenosine deaminase (ADA) [45]. Inosine is further metabolized via a series of enzymes to the end-product of adenosine metabolism, uric acid. Alternatively, adenosine can be generated from the sequential breakdown of ATP [46]. ATP can be released from the cell into the extracellular space via several different mechanisms including vesicular release, lysosomes, cell lysis, and nucleotide-permeable channels [47,48,49,50,51]. ATP is then rapidly catabolized to adenosine by the extracellular enzymes ectonucleoside triphosphate diphosphohydrolases (ENTPD 1-3, 8; Table 2) [52] which hydrolyze ATP to ADP/AMP. Ecto-5′nucleotidase (NT5E; Table 2) is the primary rate-limiting enzyme that converts AMP to adenosine [53,54,55]. During pathophysiological states, ATP and adenosine are released into the extracellular space to fine-tune the immune system [56]. ENTPD1 and NT5E play significant roles in the regulation of immunity and inflammation [57,58]. This enzyme cascade modulates purinergic signaling by driving a shift from a pro-inflammatory ATP state to an adenosine-induced anti-inflammatory environment, a role that is facilitated by their widespread expression on almost all tissue cell types [59]. Thus, adenosine and ATP levels are closely interrelated and the ratio of adenosine to ATP acts as a significant modulator of their biological effects.

## 4. The Adenosine System: Purinergic Receptors

Adenosine and ATP activate two main families of receptors; the purinergic P1 and P2 receptors; respectively (see Appendix A for receptor nomenclature; ligands; G-protein; and ions) [60,61,62,63,64,65,66,67,68]. The G-protein coupled P1 adenosine receptors are inhibitory (A_1_ and A_3_) and excitatory (A_2A_R and A_2B_R); with high affinity (A_1_R and A_2A_R) (0.1–0.3 μM) and low affinity (A_2B_R; A_3_R) (15–25 μM) for adenosine [36,69,70], although some studies using functional assays report that adenosine is approximately equipotent at its receptors [71,72,73]. Purinergic receptor activation is a tightly regulated process. A_1_Rs are typically tonically activated by endogenous extracellular adenosine whereas A_2A_Rs are selectively recruited by adenosine produced through extracellular catabolism of ATP by ectonucleotidases [74,75,76,77].

Due to the widespread constitutive expression of purinergic receptors in the body and the ability of P1Rs to form homo- and heterodimers with other neurotransmitter receptor types [68], adenosine and ATP serve significant roles in modulating neural processes, immune response, energy metabolism, and sleep [69,70,71]. Presynaptic A_1_ receptor activation inhibits excitatory synaptic transmission by reducing calcium influx and glutamate release, while post-synaptic A_1_ receptor activation reduces ionotropic glutamate receptor and voltage-sensitive calcium channel activation [78] leading to a decrease in excitatory synapse activity. Conversely, the actions of facilitatory A_2A_Rs increase the release of glutamate and ionotropic glutamate receptor function in different brain regions. Their expression on astrocytes regulates glutamate uptake and Na^+^/K^+^ ATPase, contributing to the role of A_2A_Rs as selective mediators of synaptic plasticity [78]. Increasing the complexity of adenosine’s neuromodulatory action [79,80], adenosine receptors can also form heterodimers with other G-protein coupled receptors [81]. Activation of antagonistic A_2A_-D_2_ receptor heteromers, first identified in striatal membrane preparations [82] and found in GABAergic striatopallidal neurons [83], results in reduced D_2_ receptor affinity for agonists. A_2A_-D_2_ receptors expressed on striatal astrocytes also functionally interact to modulate glutamate gliotransmitter release [84]. Inhibitory A_1_ receptors dimerize with excitatory dopamine D_1_ receptors to form receptor heteromers that modulate striatonigral neuronal function [83]. Adenosine’s ability to modulate glutamatergic and dopaminergic pathways is of relevance in the pathophysiology of schizophrenia.

ATP and adenosine also regulate immune responses via activation of purinergic receptors, leading to immune cell activation, signal amplification, chemotaxis, and phagocytosis [56,85,86,87]. ATP acts as a pro-inflammatory danger associated molecular pattern (DAMP) when it is increased in extracellular fluids, usually indicating inflammation, ischemia, or injury [36,88]. In contrast, adenosine displays potent anti-inflammatory effects [89,90,91]. Modulation of purinergic receptors and ectonucleotidases allows for complex feedback regulation. During periods of injury or stress, cells respond by increasing ATP release into the extracellular space, inducing a pro-inflammatory response upon binding to P2 receptors. Most immune cells express several types of P2XRs and P2YRs [92]. Injured or stressed cells release ATP to recruit immune cells for phagocytosis and clearance. Immune cells can also release ATP to amplify cellular activation and regulate chemotaxis [93]. Prolonged states of inflammation can be toxic to tissues and cells. Thus, to prevent excessive inflammation and tissue damage, ATP is hydrolyzed by ectonucleotidases to generate adenosine. The decreased extracellular ratio of ATP:adenosine reduces P2 signaling and increases P1 signaling, specifically the upregulation and activation of A_2A_Rs and A_2B_Rs. Engagement of A_2A_Rs allows for the inhibition of lymphocyte activation [94] while activation of A_2B_R increases the production of vascular endothelial growth factor (VEGF) and interleukin-6 (IL-6) [95]. A_2B_Rs, which are mainly expressed on macrophages, dendritic cells, and mast cells, may then contribute to the resolution of inflammation by promoting healing [96]. The endogenous anti-inflammatory role of adenosine is utilized in therapies for acute and chronic inflammatory diseases like chronic obstructive pulmonary disease and rheumatoid arthritis [97,98].

## 5. The Adenosine System: Perturbed Adenosine Metabolism in Disease: Uric Acid

The final product of adenosine metabolism, uric acid, is used clinically to assess purine catabolism. Altered levels of uric acid in the blood of schizophrenia patients implicates impairment of the adenosine system in the pathophysiology of this disorder. Reduced plasma uric acid levels are reported in clinically stable and relapsed schizophrenia patients [99] and in first-episode neuroleptic-naïve schizophrenia patients [100] compared to healthy controls. Although schizophrenia patients are not hypouricemic (generally defined as <2 mg/dL), they report lower uric acid plasma levels (schizophrenia male—5.1 mg/dL, schizophrenia female—4.4 mg/dL; healthy control male—6 mg/dL, healthy control female—5.1 mg/dL), suggesting perturbed purinergic signaling [100]. In contrast, higher levels of uric acid are reported during the acute phase in schizophrenia patients [101]. Similarly, during psychosis relapse, elevated uric acid levels have been found in schizophrenia patients that were as high as in bipolar disorder patients during manic episodes and higher than in the healthy control group [102]. These inconsistent findings of uric acid levels may be attributed to the relatively small study sizes, differences in uric acid levels at different stages of this heterogeneous disorder, or the effects of antipsychotic medications. A recent meta-analysis of 17 studies addressed these limitations, and in subgroup analyses He et al., found that uric acid levels are significantly reduced in first episode psychosis (Weighed mean difference −40.61 μmol/L, CI_95%_ −59.47–−21.76, *p* < 0.0001) in schizophrenia patients but not in those with chronic schizophrenia (Weighted mean difference = −3.09 μmol/L, CI_95%_ −35.21–29.04, *p* = 0.85) [103]. They also reported uric acid levels are reduced in male (Weighed mean difference = −34.83 μmol/L, CI_95%_ −54.97–−14.69, *p* = 0.0007) and American schizophrenia patients (Weighed mean difference = −39.90 μmol/L, CI_95%_ −61.47–−18.33, *p* = 0.0003). These findings suggest disease-stage and sex-dependent differences in uric acid levels indicating an alteration in purine catabolism early in the course of the disease. Hypofunction of the adenosine system in schizophrenia lends one possible explanation for these decreased uric acid levels in patients, as a lowered availability of adenosine restricts uric acid production.

Uric acid also plays a dual role in the body’s antioxidant defense system [104]. Low levels of uric acid decrease the body’s ability to prevent free radical damage, leading to cellular damage and death [105]. Conversely, high levels of uric acid (hyperuricemia, generally defined as >6 mg/dL) contribute to an inflammatory response, as seen in disorders such as gout and metabolic syndrome [106,107]. Uric acid acts as a powerful antioxidant to neutralize free radicals, such as peroxynitrite and hydroxyl, making its levels in the blood a strong indicator of oxidative stress. There is robust evidence supporting an increased oxidative stress status in schizophrenia [108,109], while the use of antioxidant supplementation, like vitamin E and C, improves some of the psychopathological symptoms of schizophrenia [110,111,112]. Thus, as well as reflecting changes in adenosine metabolism, baseline levels of purine metabolites are a good predictor of clinical and neurological symptoms in schizophrenia [113].

Hyperuricemia is associated with cancer incidence and mortality [114,115]. Elevated serum uric acid (6.8 mg/dL or 404 uM without the presence of gout) is positively correlated with risk for gastrointestinal cancers and an increased overall mortality risk for cancers [116]. High serum uric acid (>6 mg/dL) is used as a prognostic predictor of colorectal cancer, pancreatic cancer, large B-cell lymphoma, and esophageal squamous cell carcinoma [117,118,119,120]. A study examining hyperuricemia in cancer development showed spontaneous hepatocellular carcinoma and hepatomegaly in mice following knockout of the enzyme, *Urah*^plt2/plt2^, responsible for uric acid metabolism [121]. These findings point to altered adenosine catabolism as contributing to cancer incidence and mortality. Adenosine levels are increased in the TME but not in plasma of patients with cancer [122,123]. Equally, expression and activity of the enzyme ADA, which metabolizes adenosine to inosine, is also increased in tumors [124], but similar increases in serum may originate from other sources [125]. Increased uric acid levels likely reflect the increased abundance of adenosine in the TME and are an indicator of oxidative stress in response to cancer.

We posit that the adenosine system is hypofunctional in patients with schizophrenia resulting in reduced uric acid generation, in line with the reports of lower peripheral uric acid levels, which potentially mitigates the increased risk for cancer associated with hyperuricemia. Hypo- and hyperuricemia are associated with disease and although animal models indicate a potential causative role for hyperuricemia in hepatic cancers [121], the etiology of dysregulated uric acid levels in these diseases is not yet known. Rather, uric acid acts as a marker of purine catabolism, suggesting significant perturbation of the adenosine system in both schizophrenia and cancer.

## 6. Schizophrenia: Evidence for Adenosine System Perturbation

Schizophrenia is a severe, debilitating mental illness characterized by paranoid delusional thinking, hallucinations and alterations in volition, neurocognition, and affective regulation. While the global incidence of schizophrenia is relatively small, estimated at 0.30–0.66%, there is a substantial burden to the individual afflicted with the disease and to society [126,127]. One evolving theory describing the etiology of schizophrenia is dysfunction of the adenosine system. As a neuromodulator of both the glutamate and dopamine neurotransmitter systems, perturbation of adenosine may reconcile the hyperdopaminergic and hypoglutamatergic states that underlie the positive, negative, and cognitive symptoms found in patients with schizophrenia. This hypofunction of adenosine hypothesis developed from observation of several phenomena: (1) presynaptic activation of adenosine A1 receptors inhibits the release of glutamate, (2) postsynaptic activation reduces NMDA-receptor functioning, and (3) adenosine signaling through postsynaptic A_2A_ receptors inhibits D_2_ receptor signaling through a heteromeric complex, as reviewed [128]. Both dysfunction of the glutamate and the dopamine systems have been attributed to the etiology of schizophrenia.

The enzymes responsible for the generation and regulation of extracellular adenosine are significantly altered in the brain in schizophrenia. Protein expression of the equilibrative nucleoside transporter (ENT1), which transports adenosine, is reduced in SCZ in the superior temporal gyrus [129]. Protein expression and enzyme activity of ENTPD1 and NT5E are significantly reduced in the striatum in schizophrenia subjects [130]. ENPTD1 and ENTPD2 transcript levels were also significantly reduced in enriched populations of astrocytes, but not neurons, in the dorsolateral prefrontal cortex (DLPFC) in schizophrenia [131]. Conversely, transcript levels of adenosine deaminase (ADA), the enzyme responsible for metabolizing adenosine to inosine, were significantly increased in an enriched population of pyramidal neurons in the same study, suggesting region- and cell-subtype specific dysregulation in adenosine metabolism. Although it has been proposed that increased expression of the enzyme adenosine kinase may drive adenosine hypofunction [39,132], recent findings show no significant changes in adenosine kinase expression in the brain in schizophrenia [133]. This suggests that hypofunction of ectonucleotidase and ADA pathways are likely responsible for the altered availability of extracellular adenosine.

Purinergic receptors are also significantly dysregulated in schizophrenia. Polymorphisms in the A_1_ receptor, but not the A_2A_ receptor, are associated with schizophrenia [128]. In the striatum, a reduction in the transcript and the protein expression of the A_2A_ receptor, attributed to hypermethylation of the A_2A_ coding region, was found in schizophrenia [134]. Increases in striatal A_2A_ receptor density [135] and caudate nucleus A_2A_ and D_2_ receptor protein expression [136] are reported in schizophrenia patients. Conversely, A_2A_-D_2_ receptor heterodimers were significantly reduced (almost 60%) in the same subjects [136]. These results lend support to the adenosine hypothesis of schizophrenia and deficits in the neuromodulatory action of adenosine on the dopamine system via the A_2A_-D_2_ receptor heteromers. No changes were found in A_1_ receptor expression in the striatum [133], but there was a reduction in the A_1_ receptor transcript in an enriched population of pyramidal neurons from the DLPFC in schizophrenia subjects [131]. Little is known about A_2B_ and A_3_ receptor expression in the brain in schizophrenia, likely due to their low levels of expression in this tissue. As a modulator of the dopamine and glutamate neurotransmitter systems, dysfunction of the adenosine system is hypothesized to contribute to the dopaminergic hyperfunction and glutamatergic hypofunction that contributes to the onset of schizophrenia symptoms.

## 7. Schizophrenia: The Adenosine System as a Therapeutic Target

The clinical treatment for schizophrenia is mainly centered on antipsychotics, specifically dopamine D_2_ receptor antagonists. However, a number of small clinical studies demonstrate the utility of the adenosine modulators allopurinol, a xanthine oxidase inhibitor [137,138], and dipyramidole, an inhibitor of the adenosine transporter (ENT) [139]. Used as add-on therapies to antipsychotics, they improved both positive (hallucinations, delusions, mania) and negative symptoms (avolition, flat affect, asociality, alogia) of schizophrenia, as assessed by the Positive and Negative Syndrome Scale (PANSS) [140,141]. However, others report no differences between adjuvant allopurinol and placebo [142]. A recent study found that administering adenosine modulators as add-on therapies significantly reduced the rate of psychiatric rehospitalization and all-cause mortality among schizophrenia patients in a Finnish cohort, particularly in younger patients (<45 years) [143]. In animal models of schizophrenia-like behaviors, augmenting adenosine levels in mice that overexpress the enzyme ADK [37] and antagonism of the P2X7R in the sub-chronic phencyclidine-induced model of schizophrenia [144], significantly improved cognitive deficits. Further investigation of the adenosine system, including therapies that directly target purinergic receptors and ATP-adenosine metabolic pathway components, may offer new avenues for the treatment of schizophrenia [145].

## 8. Cancer: Extracellular Adenosine Generation and Cancer

The TME is a multicellular complex composed of various immune cells, cancer associated fibroblasts, endothelial cells and cancer stem cells [146]. The TME suppresses immune response and supports a complex network of signals between multiple cell types that work in concert to permit and sustain tumor growth and vascularization [147,148,149]. Accumulation of extracellular ATP is observed in the TME and under hypoxic, chronic and acute inflammation conditions [122,123]. Extracellular ATP levels are low (nanomolar range) in healthy tissue however upon cellular stress, damage or cell death, extracellular ATP can increase significantly (micromolar range) [123,150]. As ATP is released from cancer cells, it is metabolized to immunosuppressive AMP and adenosine, which act as key modulators of immune cells [36]. As in the brain, adenosine in the TME is primarily produced by extracellular metabolism of ATP via a cascade of ectonucleotidase enzymes (Figure 2) [151]. Studies examining ex vivo adenosine levels in rodent models found adenosine to be twice as high in tumors as compared to other tissues and at least 30% higher in the tumor core [123]. The increased presence of adenosine in the TME serves to regulate cancer cell growth and immune cell activity following adenosine receptor activation [123,152,153].

Enzymatic pathways that regulate the production of extracellular adenosine are also of importance in cancer development. The purine salvage pathway enzyme, ADA, modulates ATP:adenosine levels. ADA activity is increased in breast, kidney, and colorectal cancers as well as lymphocytes, a potential compensatory mechanism to offset elevated adenosine production [125,154,155,156]. Conversely, ADA activity was decreased in prostate and gastric tumors, as well as Hodgkin’s lymphoma [157,158,159]. This highlights the dynamic metabolomic demands of cancer tissues and perturbation of adenosine metabolic processes in cancer.

Increased production of extracellular adenosine promotes both cancer cell proliferation and suppression of immune cells in the TME. There is abundant evidence that ENTPD1 and NT5E expression are increased in cancer including in gastric cancer, non-small cell lung cancer and prostate cancer [160,161,162,163]. ENTPD1 is elevated in intratumoral immune cells of non-small cell lung cancer [164]. Adult T-Cell leukemia/lymphoma cells that express high levels of ENTPD1 effectively evade antitumor immunity [165]. Similar findings have been reported in human follicular lymphoma, glioblastoma multiforme, breast cancer, rectal adenocarcinoma, non-small cell lung cancer, head and neck cancer, and medulloblastoma [166,167,168,169,170,171,172]. In immunocompetent rats, a NT5E siRNA was used to assess NT5E contribution to immune cell evasion and tumor growth in glioblastoma. In this model, decreased T lymphocyte infiltration and tumor cell apoptosis was observed [173]. Similarly, knockdown of ENTPD1 (CD39) in cancer cells and in mice yielded lower ATP consumption and increased T-Cell infiltration in tumors [174]. Additionally, a meta-analysis of 13 studies analyzing the prognostic value of NT5E, indicates NT5E expression correlates with poor overall survival and disease-free survival in solid-tumor cancer patients [175]. ENTPD1 and NT5E, as important regulators of extracellular adenosine availability, facilitate evasion of the immune system by cancer cells. These findings illustrate widespread upregulation of ATP-adenosine levels and metabolism by ectonucleotidases, resulting in cancer cell proliferation and immunosuppression in cancer. Targeting elevated adenosine levels and ATP-adenosine metabolism, may increase immune response in cancer.

Extracellular adenosine binds to the A_2A_ receptor expressed on macrophages, monocytes, dendritic cells, natural killer cells and T-Cells [41,123,176,177,178] and the A_2B_ receptor on macrophages and dendritic cells [96,179]. This increases intracellular cAMP leading to cAMP/PKA signaling that suppresses T-cell response [180]. However, adenosine receptor activation has pleiotropic effects on other immune cells. It prevents dendritic cell activation but also increases the release of VEGF, a signaling protein that promotes angiogenesis thereby promoting tumor growth [96,181]. In tissues where adenosine concentrations are low, adenosine activation of A_1_Rs can enhance neutrophil activity [85]. Adenosine can also induce chemotaxis in immature dendritic cells via A_1_R activation [182], whereas mature dendritic cells are sensitive to adenosine-mediated inhibition through the A_1_ receptor [181]. Overall, adenosine receptor activation has an immunosuppressive effect on different mature (and immature) immune cells and aids the progression of cancer by facilitating immune-surveillance evasion.

All adenosine receptors subtypes seem to contribute to cancer cell proliferation via activation of different kinase signaling pathways including AKT, ERK1/2, p38, JNK and PKC-δ pathways [42,183,184,185,186,187,188,189]. Colon cancer cells, where A_2A_ and A_3_ are the most abundantly expressed receptor subtypes, undergo increased cell proliferation when treated with high (micromolar) concentrations of adenosine [189]. The A_2A_R contributes to angiogenesis and wound healing indicating that the downstream targets of adenosine receptors, including ERK, JNK, p38, and AKT, contribute to tumor growth via multiple signaling pathways [190,191]. Bladder urothelial carcinoma tissues and cells express A_2B_R at higher levels than other adenosine receptor subtypes. High expression of A_2B_R is correlated with poor prognosis, and suppression of A_2B_R inhibited proliferation, invasion, and migration of bladder urothelial carcinoma cells with arrest at the G1 phase [188]. Oral squamous cell carcinoma cells (OSCC) are inhibited following knockdown of A_2B_R. Interestingly, the P2 purinergic receptor P2X7, which is expressed on tumor-associated immune cells, also contributes to immunosuppression of cancer in the tumor microenvironment [192,193,194,195]. Additionally, knockdown of the P2X7 receptor suppresses TGFβ-1 induced cell migration in A549 lung cancer cells and actin remodeling, illustrating that activation of P2X7 contributes to cell migration [196].

In summary, purinergic signaling is important to cancer cell progression and immune response through a variety of pathways. Elevated extracellular ATP acts on P2 receptors, which are abundantly expressed in many cancer subtypes and immune cells which contribute to increased cancer cell proliferation and immune evasion. ATP is metabolized to adenosine which acts on P1 receptors, whose immunosuppressive role similarly helps tumors evade immune cells.

## 9. Cancer: Adenosine System as a Therapeutic Target in Cancer

A number of different anti-cancer therapies that target the adenosine system are currently in development (Table 3). Small molecules (See drug formulas Table 3 and structures in Appendix A) have been developed to target the adenosine receptors to improve cancer immune response and inhibit proliferation and metastasis of cancer cells. A_2A_ receptor antagonists currently in clinical trials include compounds CPI-444, MK-3814, NIR178, and AZD4635 that are administered as monotherapies or in combination with PD-1 and PD-L1 antagonists [197]. A_2A_R antagonists like AZD-4635, CPI-444, and NIR-178 are under investigation for their ability to improve T-Cell infiltration in tumors. Lesser studied A_2B_ and A_3_ receptor antagonists, like PBF-1129 and CF-102, respectively, are also undergoing clinical trials (Table 3, Figure 1).

A promising but relatively understudied strategy is targeting adenosine-generating enzymes. Due to their higher specificity and longer half-life, monoclonal antibody-directed chemotherapies are used to target the adenosine generating enzyme NT5E (see Table 3) [198,199]. Internalization of NT5E following treatment with monoclonal antibody therapies prevents adenosine-mediated action on tumors, as well as preventing metastasis [199]. Therapies targeting ENTPD1 are predominately still exploratory. By reducing adenosine availability in the extracellular space, ENTPD1-targeted therapies seek to prevent metastasis and improve cancer immune response by restoring T-reg function [200,201,202]. In breast cancer, reducing ENTPD1 activity and adenosine generation prevents A_2B_ receptor activation, reducing metastasis-inducing transcription factor FRA1/FOSL1 expression [203].

Adenosine system targeting cancer therapies currently being investigated in clinical trials highlight the importance of elucidating the role of adenosine system dysfunction in the development of novel cancer therapeutics.

## 10. Summary and Conclusions

One of the most surprising findings from epidemiological studies of schizophrenia is the reportedly low incidence of cancer relative to the general population. Conflicting reports of cancer rates in schizophrenia have been published for decades, even though this population are considered at increased risk of developing cancer, due to the prevalence of risk factors like smoking and access to medical care [4]. Such controversial findings lend support for the idea that understanding the pathophysiology of schizophrenia may offer unique insight into the etiology of cancer.

Overall, there is a paucity of data available on cancer-related markers in schizophrenia patients who develop cancer. The available literature on this unique population has breadth but lacks depth, as the publicly available epidemiological data sets are limited to diagnoses and procedures (e.g., National Inpatient Sample). Several studies consider genetic risk variants that may be relevant to schizophrenia and cancer, but there is little data available from clinical laboratory tests reporting on biological markers for cancer in schizophrenia patients [204,205,206]. Thus, we consider reports from postmortem and clinical studies that assess the purinergic system in these disorders.

The adenosine system is hyperfunctional in cancer, facilitating tumor growth, survival and proliferation while evidence suggests this system is hypofunctional in schizophrenia. While it is challenging to determine if hypofunction of the adenosine system in the brain in schizophrenia extends systemically (and may impact cancer development throughout the body), corresponding changes in peripheral uric acid levels in a subset of the schizophrenia population suggest that this may be the case. Perturbation of ATP and adenosine metabolism, resulting in reduced availability of extracellular adenosine and dysregulated purinergic receptor activation, may attenuate the pathological oncogenic processes driven by elevated purine levels. Adenosine and ATP contribute to tumorigenesis through P1 and P2 receptor mediated signaling. Importantly, several P1 receptor antagonists are under investigation as therapies for different cancers. In schizophrenia, modulating adenosine catabolism, rather than directly targeting P1 receptors (due to associated side effects) is a moderately successful strategy at alleviating symptoms of schizophrenia. This approach is also used in cancer treatment, and drugs targeting ENTPD and NT5E are under exploration.

Much work is required to determine why and how a schizophrenia diagnosis appears to protect against the development of cancer. Schizophrenia and cancer are complex, heterogeneous disorders; multiple genetic and environmental factors interact leading to the onset of these disorders. Perturbation of neurobiological processes, like the adenosine system, may underlie the lower incidence of cancer reported in some schizophrenia patients, and lends support for targeting this system therapeutically in these disorders.

## Figures and Tables

**Figure 1 ijms-23-11835-f001:**
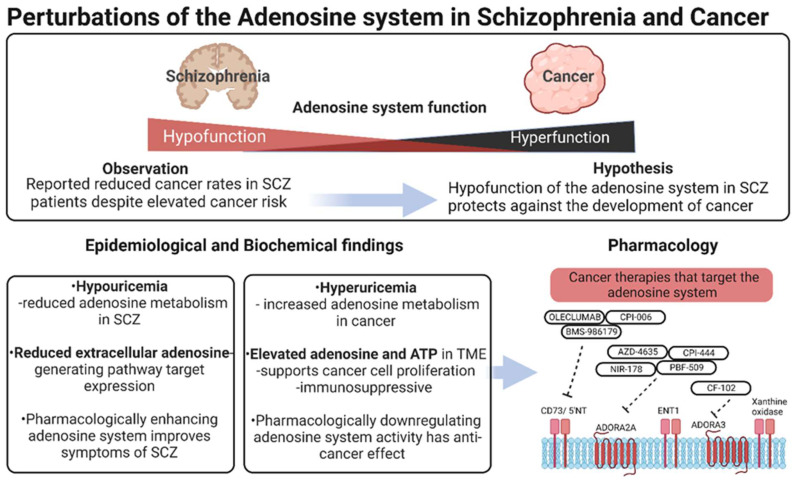
Perturbations of the adenosine system in schizophrenia and cancer. Overview of the population and biochemical evidence for perturbed adenosine system function in schizophrenia and cancer. Reduced cancer rates in schizophrenia patients despite elevated cancer risk implies some underlying biochemical protective mechanism. Reduced extracellular adenosine availability, and hypouricemia observed in these patients may explain the reduced cancer rates. Generated in Biorender.

**Figure 2 ijms-23-11835-f002:**
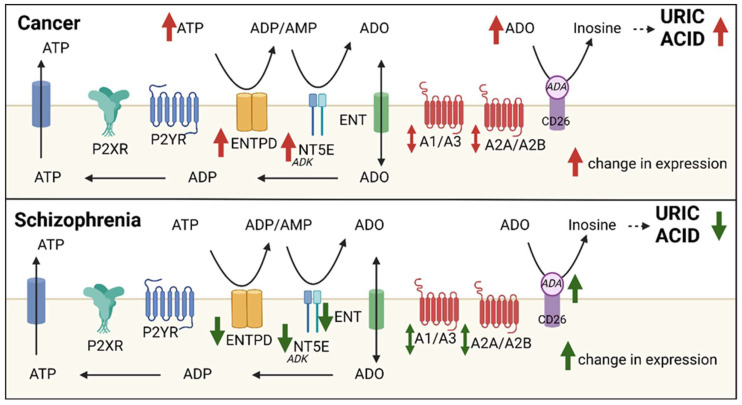
ATP and adenosine metabolism and signaling in cancer and schizophrenia. Perturbations in extracellular adenosine generating pathways are implicated in schizophrenia (green arrows) and cancer (red arrows). Differential purinergic (P1 and P2 receptors) receptor expression is reported in schizophrenia and cancer. ADA adenosine deaminase, ADK adenosine kinase, ADO adenosine, ATP adenosine triphosphate, ADP adenosine diphosphate AMP adenosine monophosphate, ENTPD ectonucleoside triphosphate diphosphohydrolases, NT5E ecto-5′nucleotidase. Generated in Biorender.

**Table 1 ijms-23-11835-t001:** Review of Meta-analyses examining cancer rates in schizophrenia patients.

Source	Sex	Participants (N)	Cancers	Major Findings
[5]	M/F	480,356 SCZ patients	All sites	Decreased overall cancer incidence (M/F) No change in females alone Sex and cancer type were confounding factors
[4]	M/F	279,938 patients across all 21 analyses	All sites	Colon, skin, and prostate cancer incidence decreased Pooled overall cancer rates were not significantly changed prior to adjusting for smoking Lung cancer rates were decreased following adjustment Breast cancer incidence was increased in female SCZ patients SCZ relatives’ cancer risk was decreased
[26]	M/F	31 studies with a median of 33,372 psychotic patients	All sites	Cancer and sex specific incidence rates were reported Risk ratios increased for oesophageal, breast, testicular, cervical and endometrial cancers Risk ratios decreased for prostate, colon, skin, and thyroid cancers Authors note better study design, controlling for confounders and appropriate comparison groups are needed
[27]	M	218,076 men across 13 studies	Prostate	Decreased risk of prostate cancer across 13 cohort studies in SCZ men (SIR = 0.61).
[11]	M/F	496,265 SCZ patients across 12 studies	Lung	No changes observed in lung cancer incidence Authors note only one study accounted for smoking and seven studies examined lung cancer incidence only after SCZ diagnosis while the rest did not
[10]	M/F	312,834 SCZ patients across seven studies	Liver	Significantly lower liver cancer incidence was observed in SCZ males (SIR—0.71) but not in females (SIR—0.83). SCZ patients have ~20% decreased risk of liver cancer Authors note confounders were not adjusted for
[23]	F	466,244 patients across 15 studies	Breast	Breast cancer incidence was increased in SCZ patients. Authors suggest morbidity (T2D, hyperprolactinemia, etc.) in female SCZ patients may contribute to higher breast cancer rates

M-Male, F-Female, SCZ-Schizophrenia, T2D-Type II Diabetes, SIR-Standardized Incidence Ratio.

**Table 2 ijms-23-11835-t002:** Nomenclature of ENTPD family enzymes.

Gene Name	Protein Name	Additional Names
ENTPD1	NTPDase1	CD39, ATPase, ecto-apyrase
ENTPD2	NTPDase2	CD39L1, ecto-ATPase
ENTPD3	NTPDase3	CD39L3
ENTPD8	NTPDase8	liver canalicular ecto-ATPase
NT5E	Ecto-5′nucleotidase	5′NT, CD73, NT5E

ENTPD/NTPDase: Ectonucleoside triphosphate diphosphohydrolase, CD39: Cluster of Differentiation 39, adapted from [54,55].

**Table 3 ijms-23-11835-t003:** Drugs targeting the adenosine system in clinical trials for cancer and schizophrenia.

Target	Drug Name	Phase	Indication	Case Number	Formulas	Combination Therapy
NT5E	BMS-986179	I, II	Advanced solid tumors	NCT02754141	Monocolonal Ab	Nivolumab, rHuPH20
	CPI-006	I, Ib	Advanced solid tumors	NCT03454451	Monocolonal Ab	Ciforadenant, Pembrolizumab
	MEDI-9447 (OLECLUMAB)	I, II	Advanced solid tumors	NCT03611556	Monocolonal Ab	durvalumab, gemcitabine, nab-paclitaxel, oxaliplatin, leucovorin, 5-FU
		I, II	Advanced solid tumors	NCT03381274		Osimertinib, AZD4635
		I, II	Advanced solid tumors	NCT03616886		Durvalumab, Carboplatin, Paclitaxel
		II	Advanced solid tumors	NCT03267589		Durbalumab, Tremelilumab, MEDI0562
		II	Breast cancer	NCT03875573		Durvalumab, Oleclumab
		I, II	TNBC	NCT03742102		Durvalumab, Capivasertib, Oleclumab, Paclitaxel, Trastuzumab deruxtecan
		II	NCSLC	NCT03334617		Durvalumab, AZD9150, AZD6738, Vistusertib, Olaparib, Oleclumab, Trastuzumab deruxtecan, cediranib
A2AR	AZD-4635	II	Prostate, mCRPC	NCT04089553	C15H11ClFN5	Oleclumab, Durvalumab
			NCSLC	NCT03381274		Osimertinib, MEDI9447
	CPI-444	I, II	NSCLC	NCT03337698	C20H21N7O3	Atezolizumab, Cobimetinib, RO6958688, Pemetrexed, Carboplatin, Linagliptin, Tocilizumab, Ipatasertib, Idasanutlin
	NIR-178	I, II	Solid tumors	NCT03207867	C10H8BrN7	PDR001
		I, II	NCSLC	NCT02403193		
	Caffeine	N/A	Schizophrenia	NCT02832401	C8H10N4O2	N/A
	Pentoxifylline	I	Schizophrenia	NCT04094207	C13H18N4O3	N/A
A3R	CF-102	Complete	Hepatocellular Carcinoma	NCT00790218	C18H18ClIN6O4	Cl-IB-MECA
		II	NASH	NCT02927314		Placebo
		I, II	Hepatitis C	NCT00790673		Placebo
		II	Hepatocellular Carcinoma	NCT02128958		Placebo

mCRPC Metastatic Castration-Resistant Prostate Cancer, NASH Non-alcoholic steatohepatitis NCSLC Non-Small Cell Lung Cancer, TNBC Triple-Negative Breast Cancer.

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
