# Peer review of "Adenosine, Schizophrenia and Cancer: Does the Purinergic System Offer a Pathway to Treatment?"

_ijms, 2022, doi:10.3390/ijms231911835_

Round 1

Reviewer 1 Report

In the present review manuscript, Hamoud et al try to establish a link between adenosine, schizophrenia, and Cancer.

A serious concern about this manuscript is that it promotes the speculation that patients with Schizophrenia are somehow protected against cancer.

A serious review of the literature shows that the incidence of cancer in people with schizophrenia has been reported to be increased, reduced, or similar compared to the general population. The reported lower incidence rates in some studies have fueled speculation that schizophrenia is protective against cancer development. However, note that this has not been conclusively validated and we do not know anything about biological and other mechanisms underlying this.

Note that

1.      Patients with Schizophrenia have a 20% lower life span than the general population making a comparison cancer that occurs in late stages of life incidence difficult to interpret.

2.      Patients diagnosed with schizophrenia have an approximately 50% increased risk of death by cancer compared to age and sex-matched people in the general population therefore the idea that schizophrenia protects against cancer is not correct.

3.      The antipsychotic drugs that are being repurposed for cancer therapies are being investigated due to their ability to promote autophagy, apoptosis and/or cell cycle arrest. This is entirely independent of their anti-schizophrenic action on relevant brain structures or neurotransmitter systems.

Additionally, the studies that report an increased incidence are extremely cautious in their description. However, this manuscript bluntly states that schizophrenia protects against cancer.

For this reason, authors are advised to change the narrative in the manuscript.

Additional comments-

1.      Lines 17-18 (Abstract): The statement- the adenosine system is seen as an obstacle to the development of effective immune-targeted therapies is not supported. Perhaps a dedicated paragraph for adenosine and cancer immunotherapy would help clarify and make the point.

2.      Page 6 lines 207-210 as well as page 7 lines 250-253. Authors include contradictory results that are difficult to interpret in the present context or at least not in the way authors have presented.

Author Response

Response to Reviewers

We thank the reviewers for their thoughtful comments and the opportunity to improve the manuscript. We have addressed the concerns raised by each reviewer below. Changes made to the text are highlighted in yellow in the manuscript.

Reviewer 1:

In the present review manuscript, Hamoud et al try to establish a link between adenosine, schizophrenia, and Cancer. A serious concern about this manuscript is that it promotes the speculation that patients with Schizophrenia are somehow protected against cancer.

A serious review of the literature shows that the incidence of cancer in people with schizophrenia has been reported to be increased, reduced, or similar compared to the general population. The reported lower incidence rates in some studies have fueled speculation that schizophrenia is protective against cancer development. However, note that this has not been conclusively validated and we do not know anything about biological and other mechanisms underlying this.

Response: We thank the reviewer for their important feedback. As the reviewer suggests, in the Introduction section “Cancer incidence in schizophrenia” we discuss the conflicting findings in the literature of increased, decreased and no change in cancer incidence in schizophrenia patients. We also refer the readers to several sources that discuss the challenges of interpreting cancer incidence rates in this patient population. We highlight that, as other groups have reported (Table 1), it is this conflict in the literature, rather than low cancer incidence reports per se, that suggests that some (not all) cancer rates may be lower in schizophrenia patients.

Note that

Patients with Schizophrenia have a 20% lower life span than the general population making a comparison cancer that occurs in late stages of life incidence difficult to interpret.

Response: The reviewer is correct. We note this as a limitation of the schizophrenia-cancer incidence studies and include the epidemiological studies of cancer in schizophrenia patients that account for age, see “Cancer incidence in schizophrenia” and Table 1 (see highlighted section)

Patients diagnosed with schizophrenia have an approximately 50% increased risk of death by cancer compared to age and sex-matched people in the general population therefore the idea that schizophrenia protects against cancer is not correct.

Response: We thank the reviewer for this important comment. We now clarify that we are discussing cancer incidence rates and not cancer mortality rates, which as the reviewer correctly states is increased in schizophrenia patients. “Cancer incidence in schizophrenia” (see highlighted section)

The antipsychotic drugs that are being repurposed for cancer therapies are being investigated due to their ability to promote autophagy, apoptosis and/or cell cycle arrest. This is entirely independent of their anti-schizophrenic action on relevant brain structures or neurotransmitter systems.

Response: We thank the reviewer for their comment. We clarify that the expected anticancer effects of antipsychotic drugs are poorly understood (Word doc – line 109, see highlights)

Additionally, the studies that report an increased incidence are extremely cautious in their description. However, this manuscript bluntly states that schizophrenia protects against cancer. For this reason, authors are advised to change the narrative in the manuscript.

Response: We thank the reviewer for their comment. As suggested, we have edited the manuscript throughout (see highlighted sections) to discuss in greater depth the limitations of the available epidemiological studies in this patient population.

Additional comments-

Lines 17-18 (Abstract): The statement- the adenosine system is seen as an obstacle to the development of effective immune-targeted therapies is not supported. Perhaps a dedicated paragraph for adenosine and cancer immunotherapy would help clarify and make the point.

Response: We thank the reviewer for their comment. As suggested, we have edited this statement in the abstract and discuss adenosine and immune regulation (line 335, section “Cancer: Extracellular adenosine generation and cancer).

Page 6 lines 207-210 as well as page 7 lines 250-253. Authors include contradictory results that are difficult to interpret in the present context or at least not in the way authors have presented.

Response: We thank the reviewer for their feedback, the lines have been rewritten for clarity.

Reviewer 2 Report

The manuscript can be published after minor revision. In particular it would be convenient for the readers if structural formulas of discussed drugs/compounds under investigation are presented.

Author Response

Reviewer 2:

Response to Reviewers

We thank the reviewers for their thoughtful comments and the opportunity to improve the manuscript. We have addressed the concerns raised by each reviewer below. Changes made to the text are highlighted in yellow in the manuscript.

The manuscript can be published after minor revision. In particular it would be convenient for the readers if structural formulas of discussed drugs/compounds under investigation are presented.

Response: We thank the reviewer for their feedback. The drug formulas have been added to Table 3 and their structures are included in supplementary figure 1.

Reviewer 3 Report

Abstract: very much is said about why the paper is important, but quite little about the findings and conclusions. Please revise.

No keywords are present

A table with blood uric acid concentrations in controls vs. in various cancers and in schizophrenia could be very useful, as it allows at least partially evaluate the degree of hypo- or hyperfunction.

Please summarise the alterations of the adenosine system in cancer and in schizophrenia as a figure (or two separate figures) to make it easier for the reader to understand, what happens.

The incidence of cancer in schizophrenia patients deserves a separate section; please indicate the occurence rates and whether the observed differences are statistically significant.

Authors indicate that 'Adenosine system targeting cancer therapies currently being investigated in clinical trials' and describe in detail the alterations of the adenosine system in cancer. So, what are the missing pieces which author suggest to investigate or produce targeted drugs for?

Author Response

Response to Reviewers

We thank the reviewers for their thoughtful comments and the opportunity to improve the manuscript. We have addressed the concerns raised by each reviewer below. Changes made to the text are highlighted in yellow in the manuscript.

Reviewer 3:

Abstract: very much is said about why the paper is important, but quite little about the findings and conclusions. Please revise.

Response: We thank the reviewer for their feedback. As suggested, the abstract has been edited.

No keywords are present.

Response: We thank the reviewer for their comment. The keywords “Schizophrenia, Cancer, Adenosine, Purinergic Signaling, Epidemiology” have been added.

A table with blood uric acid concentrations in controls vs. in various cancers and in schizophrenia could be very useful, as it allows at least partially evaluate the degree of hypo- or hyperfunction.

Response: We thank the reviewer for this useful suggestion. The uric acid ranges reported in the relevant literature have been added to the section “The Adenosine System: Perturbed adenosine metabolism in disease: Uric acid” (see highlighted lines)

Please summarise the alterations of the adenosine system in cancer and in schizophrenia as a figure (or two separate figures) to make it easier for the reader to understand, what happens.

Response: We thank the reviewer for their feedback. As suggested, we have edited Figure 2 to more easily display alterations in the adenosine system

The incidence of cancer in schizophrenia patients deserves a separate section; please indicate the occurence rates and whether the observed differences are statistically significant.

Response: We thank the reviewer for their comment. We report the cancer incidence rates from meta-analyses of patients with schizophrenia and cancer in table 1 and discussed in the section “Cancer incidence in schizophrenia”

Authors indicate that 'Adenosine system targeting cancer therapies currently being investigated in clinical trials' and describe in detail the alterations of the adenosine system in cancer. So, what are the missing pieces which author suggest to investigate or produce targeted drugs for?

Response: We thank the reviewer for raising this interesting point. We discuss the importance of further developing therapies that target the adenosine-ATP metabolic pathways in addition to adenosine receptor targeting drugs which are under development (Section - Cancer: Adenosine system as a therapeutic target in cancer, highlighted portion, lines 423-428 in the word doc)

Reviewer 4 Report

This is a very important review. The adequate references should be overworked.

Author Response

Response to Reviewers

We thank the reviewers for their thoughtful comments and the opportunity to improve the manuscript. We have addressed the concerns raised by each reviewer below. Changes made to the text are highlighted in yellow in the manuscript.

Reviewer 4:

This is a very important review. The adequate references should be overworked.

Response: We thank the reviewer for their comments. We have added additional relevant references to the review.

Round 2

Reviewer 1 Report

The authors have addressed the previous concerns.

I only have one suggestion- Perhaps the first sentence of the abstract should read something like--- For over a century, a complex relationship between schizophrenia diagnosis and development of many cancers has been observed.